# A Systematic Review and Meta-Analysis of 29 Studies Predicting Diagnostic Accuracy of CT, MRI, PET, and USG in Detecting Extracapsular Spread in Head and Neck Cancers

**DOI:** 10.3390/cancers16081457

**Published:** 2024-04-10

**Authors:** Manish Mair, Hitesh Singhavi, Ameya Pai, Mariya Khan, Peter Conboy, Oladejo Olaleye, Rami Salha, Phil Ameerally, Ram Vaidhyanath, Pankaj Chaturvedi

**Affiliations:** 1Head and Neck Surgery, University Hospital of Leicester, Leicester LE1 5WW, UK; 2Fortis Hospital, Mumbai 400016, India; hitsinx@gmail.com (H.S.); dentistmariya@gmail.com (M.K.); 3Tata Memorial Hospital, Mumbai 400012, India; ameya1pai@gmail.com (A.P.); chaturvedi.pankaj@gmail.com (P.C.); 4Head and Neck Surgery, University Hospital of Northampton, Northampton NN1 5BD, UKp.ameerally@nhs.net (P.A.); 5Radiology Department, University Hospital of Leicester, Leicester LE1 5WW, UK

**Keywords:** ECS, neck node, head and neck cancer, imaging, USG, PET, CT, MRI, sensitivity, specificity, accuracy

## Abstract

**Simple Summary:**

In this study, we investigated the effectiveness of different imaging techniques, including CT, PET–CT, MRI, and USG, in detecting extracapsular spread (ECS) in advanced Head and Neck Cancer (HNC). The presence of tumor cells beyond lymph nodes, known as extracapsular spread (ECS), worsens the condition. This research compares different imaging techniques like CT, MRI, PET–CT, and USG to detect ECS before surgery. The findings show that while MRI and PET–CT are more sensitive than CT in spotting ECS, all methods have similar accuracy in correctly identifying it. This suggests the need to refine the criteria used in imaging to better diagnose ECS in HNC, which could significantly impact how this cancer is treated in the future.

**Abstract:**

Background: Extracapsular spread (ECS) is the extension of cancer cells beyond the lymph node capsule and is a significant prognostic factor in head and neck cancers. This meta-analysis compared the diagnostic accuracy of CT, MRI, PET, and USG in detecting ECS in head and neck cancers. Methodology: The authors conducted a systematic review and meta-analysis of studies that compared the diagnostic accuracy of CT, MRI, PET, and USG in detecting ECS in head and neck cancers. They included studies that were published between 1990 and December 2023 and that used histopathology as the reference standard for ECS. Results: The pooled sensitivity and specificity of CT scan were 0.63 (95% CI = 0.53–0.73) and 0.85 (95% CI = 0.74–0.91), respectively. The pooled sensitivity and specificity of MRI were 0.83 (95% CI = 0.71–0.90) and 0.85 (95% CI = 0.73–0.92), respectively. The pooled sensitivity and specificity of PET were 0.80 (95% CI = 0.74–0.85) and 0.93 (95% CI = 0.92–0.94), respectively. The pooled sensitivity and specificity of USG were 0.80 (95% CI = 0.68–0.88) and 0.84 (95% CI = 0.74–0.91), respectively. MRI had significantly higher sensitivity than CT scan (*p*-0.05). The specificity of CT and MRI was not significantly different (*p*-0.99). PET scan had the highest specificity among all imaging modalities. Conclusion: MRI is the most accurate imaging modality for detecting ECS in head and neck cancers. CT scan is a reasonable alternative, but PET scan may be considered when high specificity is required. USG may not add any further benefit in detecting ECS.

## 1. Introduction

According to Globocan 2020, 747,476 new head and neck cancer (HNC) (excluding thyroid cancers) cases were diagnosed, and unfortunately, 367,285 died due to HNC [1]. Cervical lymph node metastasis is one of the most important risk factors influencing prognosis of these HNCs [2]. The presence of extracapsular extension (ECS) in these metastatic nodes has further detrimental effect. Studies have shown that the presence of extranodal extension negatively impacts locoregional recurrence and distant metastasis. Therefore both the randomized controlled trials conducted by Cooper and Bernier advised adding concurrent chemotherapy and radiotherapy in adjuvant settings after surgery. Therefore, the presence of extranodal extension and positive margins are the two independent risk factors for concurrent chemotherapy and radiotherapy in adjuvant settings [3]. Thus, ECS results in trimodality therapy leading to treatment-related side effects, impacting the quality of life.

Extracapsular extension is often interchangeably used as extranodal/perinodal extension. It is defined as extension of cancer cells beyond the lymphatic capsule sometimes infiltrating the perinodal tissues. According to the Lewis classification of the ECS studies in oropharyngeal cancers, ECS can be categorized into four groups. Grade 0 is metastasis lying within the capsule of the lymph node, Grade 1 is metastasis within the nodal capsule with thickening of the capsule and disease reaching the edge of the capsule. Grade 2 is metastasis extending beyond the capsule of the lymph node, but the extension is less than 1 mm, while Grade 3 ECS is the metastasis extending for more than 1 mm beyond the capsule, and Grade 4 ECS is a soft tissue deposit without nodal tissue remnant. Studies have found grades of ECS is associated with poor survival [4]. The AJCC 8th edition of Head and Neck cancer has re-classified the N staging and overall staging both clinically and pathologically, based on the presence of the extracapsular spread (ECS) of the disease [5]. According to the latest AJCC edition, patients with the presence of ECS in the node less than 3 cm are classified as N2a node, while ECS if present in more than 3 cm are classified as N3b node; in both of the situations, the grade is classified as advanced-stage head and neck cancer [4]. The presence of ECS is usually confirmed on histology after surgery. Detecting ECS pre-operatively can help the clinician in prognostication and risk stratification of the disease. Contrast-enhanced computed tomography (CECT), positive emission tomography (PET–CECT), magnetic resonance imaging (MRI), and ultrasonography (USG) can help detect ECS pre-operatively. But, the literature is equivocal on their ability to correctly diagnose ECS. Therefore, to address this challenge, a meta-analysis was conducted to rigorously compare the diagnostic accuracy of these various imaging modalities for ECS detection in head and neck cancers. This approach will comprehensively analyze a multitude of studies, statistically pooling their results to provide a robust and objective assessment of each technique’s performance.

## 2. Materials and Methods

### 2.1. Search Strategy and Selection Criteria

The systematic review was not pre-registered but was carried out following a predefined protocol that was used to perform this systematic review and meta-analysis in accordance with the preferred reporting items for systematic reviews and meta-analyses (PRISMA) checklist [6]. The PubMed, EMBASE, and EMCARE databases were searched to identify studies reporting diagnostic accuracy of USG, PET, CT, and MRI in detecting extracapsular spread between December 1990 and December 2023. We used the following search terms: “extracapsular spread”, “ECS” and “neck node”, “cervical node” in conjunction with “USG or Ultrasonography”, “PET or positron emission tomography”, “CT or computed tomography”, “MRI or Magnetic resonance imaging”. Successive use of Boolean operators [NOT, AND, OR] was also employed. The references of all the studies were screened to include relevant additional publications. For meta-analysis on the prediction of imaging on the presence of ECS, the inclusion criteria were:

1. Head and neck cancer. 2. Availability of data on either of the imaging as mentioned above. 3. Histopathological status of the lymph node. The exclusion criteria involved: 1. Missing data on imaging characteristics of ECS. 2. Inaccurate definition of extracapsular extension. 3. Missing data on lymph node status.

### 2.2. Data Extraction and Analysis

The data were extracted by two authors [H.S. and M.M.] independently. The extracted data were confirmed by another author [A.P.]. The authors conducted a rigorous multistep selection process to identify relevant articles for this review. First, they screened titles and abstracts to find potential candidates based on the inclusion criteria. Next, they examined the references of these articles to uncover additional papers that might have been missed. Finally, they reviewed the full texts of all selected articles, assigning them to specific sections of the review and determining which ones met the stricter criteria for inclusion in the meta-analysis. Any disagreement between two authors was resolved through discussion with the third author. 

### 2.3. Assessment of Methodological Quality

The QUADAS-2 tool on RevMan 5.3 [Cochrane collaboration, Copenhagen, Denmark] was used for assessing the methodological quality of each included study [7]. The risk of bias was assessed using participant selection, index test, reference test, and flow and timing. The bias risk within each domain was categorized as either “low”, “high”, or “unclear” according to the evaluations categorized as “yes”, “no”, or “unclear”. For each included study, we recorded the author(s), medical institution, cohort size, study methods, key results, imaging used, and anatomical tumor site. Only studies fulfilling the defined eligibility criteria were recorded. We did not attempt to contact authors of studies with missing data.

### 2.4. Statistics

The statistical analysis employed a rigorous two-pronged approach, utilizing Stata version 12 software. Firstly, for each included study, a 2 × 2 contingency table was constructed. This table meticulously categorized true positives (TPs), false positives (FPs), false negatives (FNs), and true negatives (TNs) based on the ECS diagnosis established by the study authors’ pre-specified thresholds. This crucial step offered granular insights into the performance of each imaging modality across individual studies. 

Secondly, a meta-analysis was performed for each imaging technique (CT, MRI, USG, and PET scan) individually. This involved employing a mixed-effects logistic regression bivariate model. This powerful model takes into account both the inherent variability within each study due to sampling error and the potential differences in study design and execution between studies. This comprehensive approach ensures robust and generalizable results. Within the framework of the bivariate model, using the “metandi” commands in Stata, we meticulously estimated and reported combined sensitivity, specificity, positive likelihood ratio (PLR), negative likelihood ratio (NLR), and diagnostic odds ratio (DOR) for each imaging modality. These key metrics paint a holistic picture of each technique’s diagnostic accuracy, enabling clinicians to critically assess their strengths and limitations in ECS detection. Finally, to visually represent the diagnostic performance landscape, hierarchical summary receiver operating characteristic (HSROC) curves were generated. These elegant curves not only depict the average sensitivity and specificity estimates for each modality but also visualize the variability and uncertainty surrounding these estimates. This comprehensive graphical representation further empowers clinicians to make informed decisions regarding the most appropriate imaging approach for ECS detection in individual patients. In essence, the data analysis employed a rigorous and meticulous approach, providing clinicians with a nuanced and multifaceted understanding of the diagnostic accuracy of each imaging modality in ECS detection. This valuable knowledge ultimately supports more effective patient management and potentially improves clinical outcomes in the face of this challenging oncological landscape.

## 3. Results

Based on the PRISMA search strategy detailing the allocation process of the review, the above-described literature search produced 1684 papers. After removing duplicates, 1265 papers were included for further assessment. Of these, A comprehensive evaluation of the title and abstract resulted in the exclusion of 1092 articles; therefore, 173 articles were selected, and these articles underwent full-text assessment for eligibility criteria. Based on the inclusion criteria, 29 studies were included in the final analysis. Among these 29 papers, 1 manuscript reported diagnostic accuracy for CT and MRI, 1 for USG and MRI, 18 manuscripts reported diagnostic accuracy of CT [8,9,10,11,12,13,14,15,16,17,18,19,20,21,22,23] only, while 5 reported diagnostic accuracy of MRI [17,24,25,26,27,28,29] only. Among the remaining four manuscripts, diagnostic accuracy of PET and USG were reported in three [29,30,31] and one [32] papers. The literature search in the PRISMA flow diagram is shown in Figure 1. The studies characteristics are shown in Table 1. The reference standard was the histopathological report after neck dissection in all the studies. 

### 3.1. Methodological Quality of Included Studies

The main bias was the appropriate interval between index test and reference standard. The interval between neck dissection and imaging was not mentioned in few studies [8,9,12,13,14,16,17,19,22,23,24,25,28,31]. The second most common source of bias was patient selection [8,9,13,14,16,17,18,19,24,30,31]. The risk of bias and applicability concerns summary and graph are shown in Figure 2a (Risk of bias and applicability concerns summary: review authors’ judgements about each domain for each included study) and Figure 2b (Risk of bias and applicability concerns graph: review authors’ judgements about each domain presented as percentages across included studies), respectively. Table 1 describes a diverse array of studies that have investigated the diagnostic accuracy of various imaging modalities for cervical lymph nodes in head and neck squamous cell carcinoma (HNSCC). These studies, both retrospective and prospective, have enrolled patient populations ranging from 17 to 432 individuals, with mean ages generally in the 50s and 60s. While some studies focused solely on neck dissection specimens, others included whole-patient analyses. The majority of studies included both male and female participants, with male dominance observed in several cases. The sites evaluated varied across studies, with the most commonly included sites being oral cavity, oropharynx, hypopharynx, and larynx. HPV positivity information was inconsistently reported, with some studies focusing solely on HPV-positive patients, others combining both positive and negative cases, and some omitting this information entirely. Regarding imaging modalities, CT and MRI were the most frequently employed, occasionally used in combination with Doppler ultrasound or PET. Histological examination served as the reference standard for all studies.

### 3.2. Findings 

Table 2 gives an overview of pooled diagnostic accuracy of various imaging modalities in detecting extracapsular spread.

### 3.3. Computed Tomography

Figure 3 presents a forest plot analysis of CT scan sensitivity and specificity in detecting ECS. Individual study estimates exhibited variability, with sensitivity ranging from 0.16 to 1.00 and specificity spanning 0.54 to 0.98. This heterogeneity likely reflects differences in patient populations, scan protocols, and diagnostic thresholds employed. However, pooled analysis, depicted in Figure 4, reveals a more robust picture. CT scans demonstrated an average sensitivity of 0.63 (95% CI: 0.53–0.73), indicating that 63% of individuals with ECS will be correctly identified. Simultaneously, the pooled specificity of 0.85 (95% CI: 0.74–0.91) suggests a low false-positive rate of 15% among those without ECS. Further statistical interrogation underscores the diagnostic utility of CT scans. The diagnostic odds ratio of 10.1 signifies that a positive CT scan renders an individual over 10 times more likely to harbor ECS compared to a negative scan. This substantial odds ratio reinforces the clinical implication of a positive result. Furthermore, the positive likelihood ratio (LR+) of 4.33 indicates that a positive CT scan strengthens the pre-test probability of ECS by 4.33-fold, providing valuable confirmatory evidence. Conversely, the negative likelihood ratio (LR−) of 0.425 implies that a negative scan reduces the pre-test probability by nearly 60%, offering significant reassurance in such cases.

### 3.4. Magnetic Resonance Imaging

Figure 5 delves into the diagnostic efficacy of MRI to detect ECS by presenting a forest plot analysis of sensitivity and specificity across individual studies. The observed heterogeneity emphasizes the influence of study populations, imaging protocols, and diagnostic criteria on MRI performance. While sensitivity estimates varied substantially, ranging from 0.43 to 0.96, most studies clustered around the upper end, suggesting a high potential for correct ECS identification. Specificity estimates displayed similar variability, spanning from 0.72 to 1.00, indicating a generally low false-positive rate. However, pooled analysis, depicted in Figure 6, unveils a more robust picture. MRI demonstrated an impressive average sensitivity of 0.83 (95% CI: 0.71–0.90), signifying that nearly 83% of ECS cases will be accurately diagnosed. This high sensitivity, coupled with the pooled specificity of 0.85 (95% CI: 0.73–0.92) translating to a low false-positive rate of 15%, positions MRI as a highly accurate diagnostic tool for ECS. Further statistical dissection reinforces the clinical utility of MRI. The diagnostic odds ratio of 29.18 implies that a positive MRI scan renders an individual almost 30 times more likely to harbor ECS compared to a negative scan. This substantial disparity further underscores the diagnostic significance of a positive result. Moreover, the positive likelihood ratio (LR+) of 5.7 indicates that a positive MRI scan strengthens the pre-test probability of ECS by nearly sixfold, providing valuable confirmatory evidence. Conversely, the negative likelihood ratio (LR−) of 0.19 implies that a negative scan reduces the pre-test probability by over 80%, offering significant reassurance in such cases.

### 3.5. Positron Emission Tomography

Figure 7 unveils the diagnostic prowess of PET scans for identifying ECS. The forest plot reveals a tight cluster of individual study estimates for sensitivity and specificity, ranging from 0.74 to 0.85 and 0.93 to 0.94 respectively. This suggests remarkable consistency in PET’s ability to correctly diagnose and rule out ECS across diverse study populations and protocols. Pooled analysis strengthens this picture. With an average sensitivity of 0.80 (95% CI: 0.74–0.85), PET effectively identifies nearly 8 in 10 ECS cases. Simultaneously, its impressive pooled specificity of 0.93 (95% CI: 0.92–0.94) translates to a mere 7% false-positive rate, minimizing unnecessary worry and investigations. Diving deeper into statistical intricacies, the diagnostic odds ratio of 57.75 paints a stark picture. Individuals with a positive PET scan are a staggering 58 times more likely to harbor ECS compared to those with a negative scan. This significant odds ratio underscores the immense diagnostic weight carried by a positive PET result. Furthermore, the positive likelihood ratio (LR+) of 12.3 signifies that a positive PET scan multiplies the pre-test probability of ECS by over 12-fold, providing robust confirmatory evidence. Conversely, the negative likelihood ratio (LR−) of 0.21 implies that a negative scan substantially reduces the pre-test probability, offering valuable reassurance in such cases. However, despite its impressive performance, PET’s inherent limitations, including radiation exposure and cost, necessitate the exploration of alternative modalities.

### 3.6. Ultrasonography

Figure 8 casts light on the diagnostic capabilities of ultrasonography (USG) in ECS detection. Similar to PET, the sensitivity and specificity estimates for USG cluster tightly, ranging from 0.68 to 0.88 and 0.74 to 0.91, respectively. This suggests relative consistency in USG performance across studies. Pooled analysis reveals an average sensitivity of 0.80 (95% CI: 0.68–0.88), indicating that USG effectively identifies 8 out of 10 ECS cases. The pooled specificity of 0.84 (95% CI: 0.74–0.91) translates to a 16% false-positive rate, which, while higher than PET, still offers valuable diagnostic information. The diagnostic odds ratio of 20.69 reinforces the clinical utility of USG. While not as dramatic as PET’s, this odds ratio still signifies a substantial increase in ECS likelihood with a positive USG result. Additionally, the LR+ of 4.83 and LR− of 0.23 further highlight the confirmatory and reassuring roles of positive and negative USG findings, respectively.

### 3.7. Comparison of CT and MRI

Overall, there was no significant difference in the diagnostic accuracy of CT and MRI (*p* value-0.27). With regards to sensitivity, MRI was significantly better than CT scan (*p* value-0.05). The specificity values of MRI and CT were similar (*p* value-0.99).

## 4. Discussion

ECS is an independent poor prognostic factor which can be used for risk stratification and treatment intensification. Clinical ECS is defined as radiological or clinical involvement of surrounding skin, muscle, or nerve by the metastatic nodal mass. Pathological ECS (pECS) is defined as extension of tumor outside the lymph node capsule. If the tumor does not breach the capsule on histology, it does not constitute pECS. The presence of ECS significantly decreases the overall survival in head and neck cancers. Two landmark clinical trials investigated the efficacy of concomitant cisplatin-enhanced radiotherapy (CERT) versus definitive radiotherapy alone in patients with high-risk locally advanced head and neck squamous cell carcinoma (LAHNSCC). This pooled analysis identified key prognostic factors and their interaction with treatment response. Extracapsular extension (ECE) and microscopically involved surgical margins emerged as the sole risk factors demonstrating a statistically significant survival benefit from CERT in both trials. This outcome suggests that these factors may be the most potent drivers of disease progression and underscore the potential for enhanced locoregional control and disease-free survival with intensified treatment [35]. A meta-analysis by Dunne et al. consisting of 2573 patients suggested that the presence of ECS in head and neck cancers negatively impacts survival, with a summarized odds ratio of 2.7 [33]. These observations, along with the analysis of ECS on an NCDB patient population which was further validated from the MSKCC and PMH database, led to its inclusion in the new AJCC (8th edition) [33]. 

Pre-operative diagnosis of ECS will aid in prognostication and risk stratification of the disease. This is especially critical in cases of oropharyngeal cancers, where the treatment of choice is organ preservation, and pathological data are not available. The presence of ECS in these cases mandates aggressive treatment protocol. The literature on the accuracy of various imaging modalities in detecting ECS in HNC is evolving, and therefore, we have performed a meta-analysis to compare diagnostic accuracy of different imaging modalities in detecting ECS. 

Multiple radiology features have been described in the literature to characterize ECS. One of the earliest features seen is the presence of indistinct nodal margins [23]. This study aimed to quantify the ability of experienced head and neck radiologists to detect extranodal spread (ENS) of HNSCC using pre-operative CT scans. Participants underwent neck dissections and pre-surgical CT imaging, which were independently reviewed by two radiologists. Sensitivity, specificity, and positive predictive value (PPV) were calculated for both nodal involvement and ECS by comparing CT findings to histological data. A total of 149 neck dissections were analyzed. Radiologists A and B demonstrated sensitivities of 66% and 80%, respectively, for ECS detection, with specificities of 91% and 90% and PPVs of 85% and 87%, respectively. The sensitivity and specificity of CT-based ECS detection in HNSCC are understudied. This study suggests moderate accuracy, with notable improvement in experienced radiologists’ hands. Another CT characteristic that is studied in the literature is breach in the capsule of the node which, when present, increases the specificity of the imaging modality in detecting ECS. Another study led by Shao Hui Huang [45] investigated the prognostic value of radiological extranodal spread (rENE) in HPV-positive head and neck squamous cell carcinoma (HNSCC) patients with positive lymph nodes (cN+). Analyzing pre-treatment scans of 517 cN+ patients, they found rENE presence (rENE+) associated with worse outcomes compared to rENE absence (rENE−): lower 5-year survival (56% vs. 85%), disease-free survival (46% vs. 83%), and locoregional control (89% vs. 96%). rENE patterns varied, with coalescent node masses being the most common. Analyzing prognostic factors, rENE emerged as the strongest predictor of both death and recurrence. They examined four patterns. Pattern 1: Tumor invasion through a single lymph node capsule but confined to surrounding fat (clear loss of sharp boundary between the capsule and fat). Pattern 2a: Tumor invasion through the nodal capsules of two separate lymph nodes. Pattern 2b: Tumor invasion through nodal capsules in two adjacent lymph nodes resulting in a coalescent node mass (most common pattern). Pattern 3: Tumor invasion beyond surrounding nodal fat planes to directly invade or encase muscles and neurovascular structures. Their analysis revealed that the presence of any rENE pattern, regardless of specific type, was associated with significantly worse outcomes compared to no rENE (rENE−). While coalescent node masses (Pattern 2b) were the most frequent type, all patterns contributed to poorer prognosis [45]. Various authors have classified the extent of radiological ECS based on the abovementioned factors [7,36,40,46]. 

We found that the sensitivity of MRI (0.83 vs. 0.63) was significantly higher than CT scan (*p*-0.05). We also found higher variability in the reported sensitivity of CT among the included studies, varying from 0.16 to 1.00. With regards to MRI, the sensitivity varied from 0.43 to 0.96. This variability is mainly based on the stringency and the diagnostic criteria used by the radiologist in detecting ECS. With regards to CT scan, one of the studies by Sharma et al. used central necrosis as the diagnostic criteria and had a sensitivity of 1.0, while the study by Karaman et al. used irregular margins as the diagnostic criteria and had a sensitivity of 0.16. With regards to MRI, Moreno et al. [28] showed a low sensitivity value of 0.43 with diagnostic criteria of loss of the adjacent fat plane. The node’s border is poorly defined, and it is difficult to distinguish it from nearby structures. This prospective study evaluated the efficacy of 3 Tesla Magnetic Resonance Imaging (MRI) in detecting extracapsular spread (ECS) of squamous cell carcinoma (SCC) of the tongue. Twenty-five patients underwent pre-operative MRI and subsequent neck dissection surgery. MRI exhibited moderate accuracy in predicting cervical lymph node metastasis (82.4%). However, ECS detection sensitivity was modest (42.8%), while specificity remained high (100%). Its accuracy warrants its use in guiding treatment decisions. Nevertheless, limitations in ECS detection sensitivity emphasize the need for further research to improve MRI’s utility in this critical aspect of tongue cancer management. Another study led by Faraji et al. [16] suggested that the specificity increased and sensitivity decreased with the addition of each additional imaging feature.

The advantages of CT scan include easy availability, affordability, and faster scans. CT scan provides good display of nodes with regards to calcification, liquefaction, and necrosis. MRI provides better soft tissue contrast; multiplanar scanning which eventually, increases its sensitivity. The difference between the specificity of CT and MRI was not significant (*p*-0.99). Moreover, there was no significant difference in overall accuracy between the two imaging modalities. These results were reflected in the previous meta-analysis by Su et al. [47] published in 2016, consisting of 15 studies and 3971 patients. MRI demonstrated superior performance, exhibiting a mean sensitivity of 0.85 (95% CI 0.80–0.89) and specificity of 0.84 (95% CI 0.77–0.90) for ECS at the node level. Notably, specific criteria such as short-axis diameter exceeding 15 mm yielded high sensitivity, while criteria like infiltration of adjacent planes and time–signal intensity curve characteristics ensured high specificity. CT, while offering good specificity (0.85), displayed lower sensitivity (0.77). PET/CT exhibited promising results with high sensitivity (0.86) and specificity (0.86). Limited data suggested potentially good performance for US, with a mean sensitivity of 0.87 and moderate specificity of 0.75. Notably, MRI displayed significantly higher sensitivity compared to CT at the node level. However, no significant differences in specificity were observed among any modalities. While some studies exhibited potential bias, and data for patient-level analysis were limited, MRI emerges as the most valuable tool for ECS detection due to its exceptional sensitivity and specificity, especially when employing specific criteria. However, they included seven studies to summarize the diagnostic accuracy of CT scan to predict ECS. However, in recent times, there have been many additional (eleven studies, 1247 patients) studies which were published in the literature analyzing the accuracy of CT scan for the same. Pooled analysis of 18 studies has shown that sensitivity of the CT scan dropped from 0.77 to 0.65, while specificity remained more or less constant (0.85 vs. 0.83). In addition, two more studies were added with regards to MRI, and the results were almost the same as compared to the previous analysis. 

We found that PETCECT has a higher specificity as compared to other imaging techniques. This may be attributed to the additional metabolic imaging which PET–CT adds to the anatomical extent of the disease. However, only three studies reported diagnostic accuracy of PET in detecting ECS, and thus, additional literature is required to understand its true accuracy. Also, any node less than 5 mm cannot be detected by PETCECT due to the limitation of its spatial resolution. Thus, PETCECT can still miss those 12% of cases of ECS which are present in nodes less than 5 mm [4]. Few studies have also taken into consideration the SUV max values to predict ECS. In this study, a total of 120 patients were enrolled and underwent pre-operative PET-CT examinations. A clear association emerged between SUV max and occult metastasis: patients with SUV max exceeding 9.7 exhibited a significantly higher rate of occult lymph node involvement compared to those with values below this threshold (*p* = 0.041). This finding underscored the potential of SUV max as a marker for subclinical nodal spread, offering valuable prognostic information. However, there was no association of SUV max value predicting ECS in neck nodes. We await more studies to give conclusive results.

Ultrasound plays a crucial role in the sonographic assessment of cervical lymph nodes, evaluating their location, size, shape, margination, internal structure (including echogenicity, echogenic hilus, calcification, and necrosis), presence of clumping (matting), and surrounding soft tissue edema. Color or power Doppler ultrasound then investigates the vascular pattern of the lymph nodes, while spectral Doppler ultrasound further quantifies blood flow velocity and vascular resistance. In spite of the widespread use of USG, only three studies were available in the literature predicting the efficacy of USG in identifying ECS. According to our pooled analysis, USG has similar sensitivity and specificity to MRI. Sensitivity is dependent on the size, shape, and form of the node, while specificity is dependent on internal structure. Defining internal structure is more operator-dependent, thus increasing the interobserver variability and decreasing the specificity [48]. However, studies have shown contrasting results in detecting occult nodes using US. [49]

Our limitation includes the inclusion of studies from a wide span of time, thus involving imaging from varying technologies. However, we mentioned the imaging characteristic for clarity.

Future impact, perspective, and problems: Standardizing imaging criteria for extracapsular spread (ECS) across modalities is vital for accurate pre-operative detection in head and neck cancers. Challenges include criteria variability, study heterogeneity, and reporting disparities. Overcoming these through rigorous standards and collaborative research is essential for enhancing diagnostic accuracy and optimizing patient outcomes.

## 5. Conclusions

This systematic review and meta-analysis investigated the diagnostic accuracy of various imaging modalities in detecting extracapsular spread (ECS) of head and neck cancers, a critical prognostic factor influencing treatment decisions. Our analysis revealed significant differences in their performance characteristics. Magnetic resonance imaging (MRI) demonstrated superior sensitivity compared to computed tomography (CT) for ECS detection (83% vs. 63%), suggesting its preferential use for pre-operative staging. Both modalities exhibited comparable specificity (approximately 85%), highlighting the potential role of CT in resource-constrained settings. Positron emission tomography–computed tomography (PET–CT) showed the highest specificity due to its combined anatomical and metabolic imaging capabilities; however, its limited availability and resolution warrant further investigation. Ultrasound, albeit operator-dependent, displayed similar sensitivity and specificity to MRI, suggesting its potential as a readily accessible option in specific clinical scenarios. These findings provide valuable clinical insights into the diagnostic landscape for ECS detection in head and neck cancers. While MRI emerges as the preferred modality for pre-operative evaluation due to its high sensitivity, CT remains a viable alternative due to its widespread availability and cost-effectiveness. Future research should evaluate the evolving role of PET–CT and optimize the accuracy of ultrasound to expand the armamentarium for accurate ECS diagnosis and ultimately improve patient management.

## Figures and Tables

**Figure 1 cancers-16-01457-f001:**
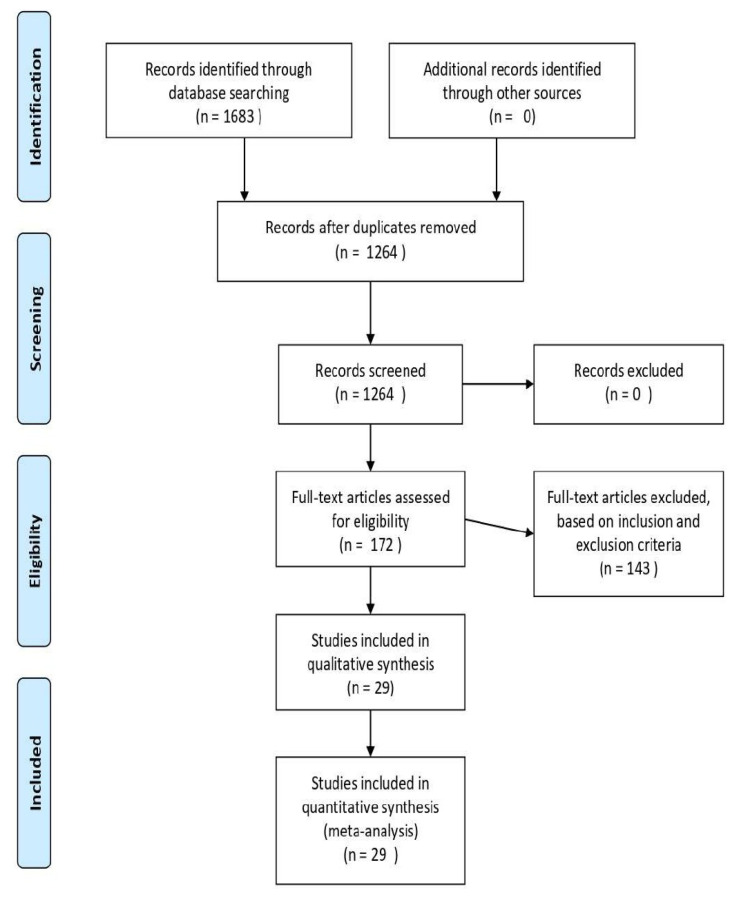
The PRISMA flow diagram.

**Figure 2 cancers-16-01457-f002:**
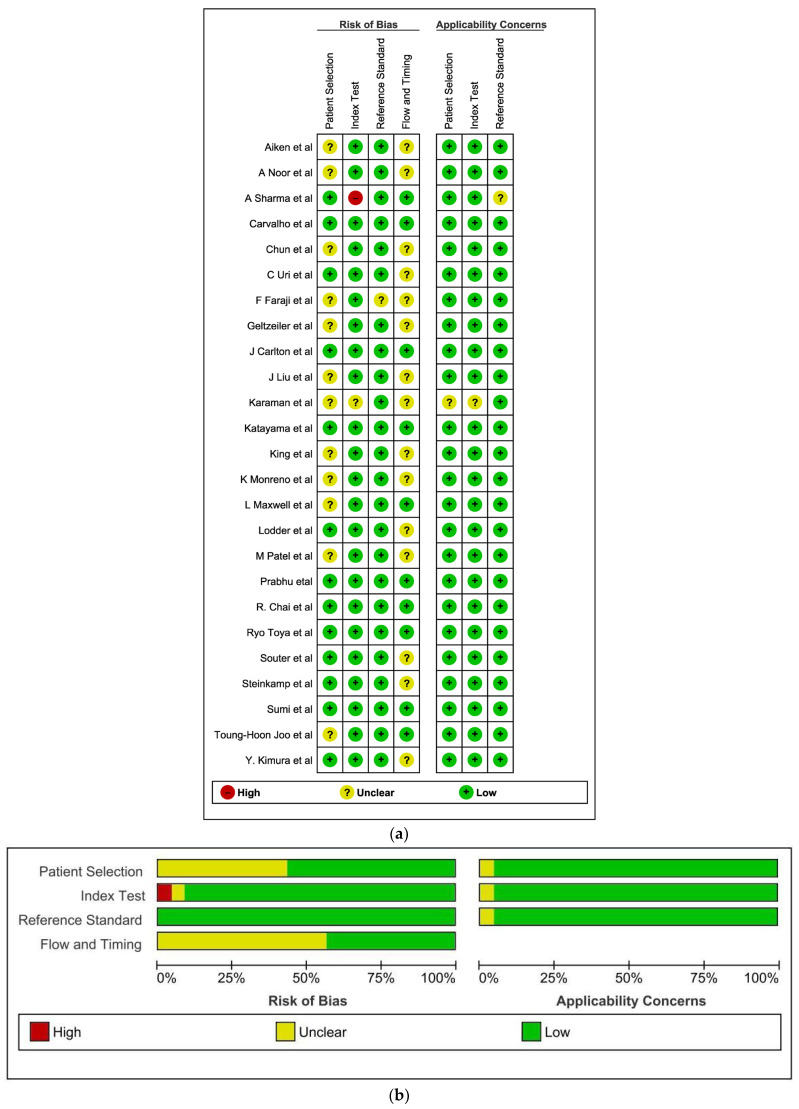
(**a**,**b**) The risk of bias and applicability concerns summary and graph [11,12,13,14,15,16,17,18,19,20,21,22,23,24,25,26,27,28,29,30,31,32,33,35,36,38,39,40,43].

**Figure 3 cancers-16-01457-f003:**
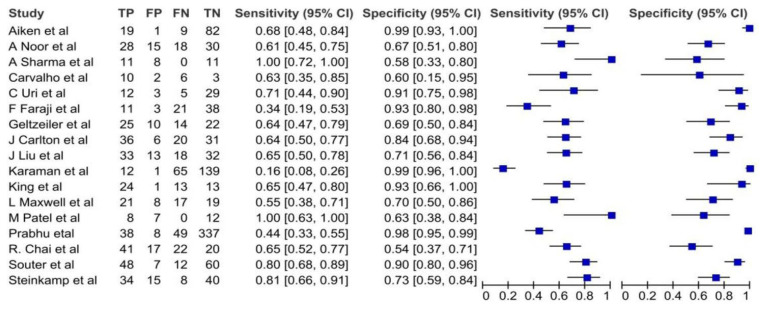
The forest plots of sensitivities and specificities of the CT scan for diagnosis of ECS [11,12,13,14,15,16,17,18,19,20,21,22,23,24,25,26,27].

**Figure 4 cancers-16-01457-f004:**
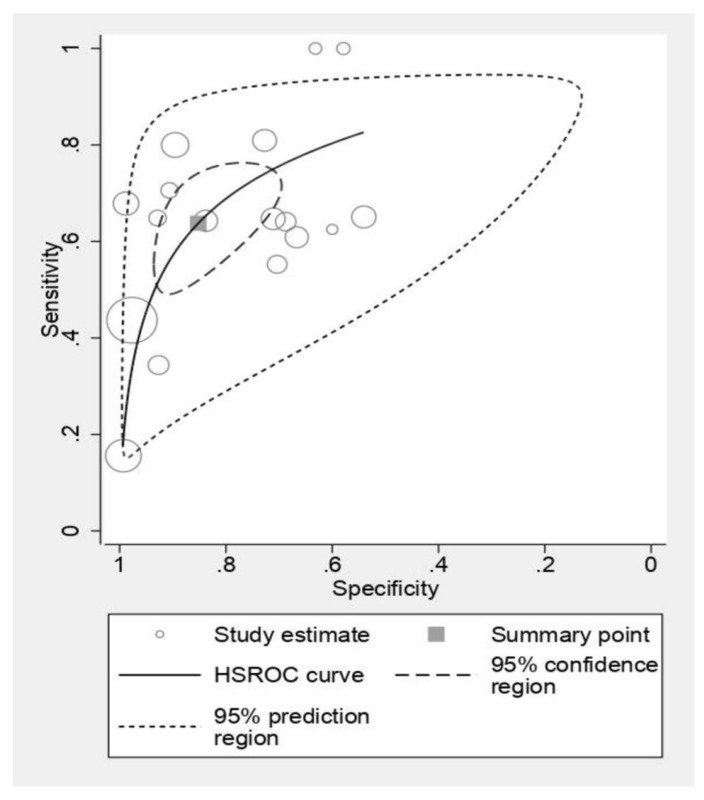
The pooled estimates of sensitivity and specificity of CT scan for diagnosis of ECS.

**Figure 5 cancers-16-01457-f005:**
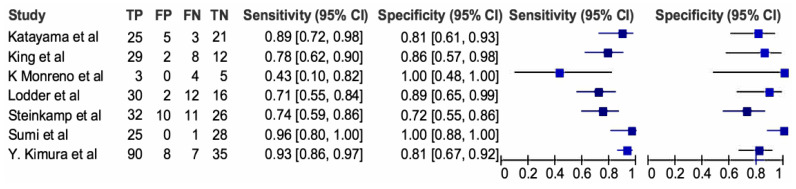
Standard summary receiver operating characteristic plot for CT scan. The forest plots of sensitivities and specificities of the MRI scan for diagnosis of ECS [20,25,28,29,30,31,35].

**Figure 6 cancers-16-01457-f006:**
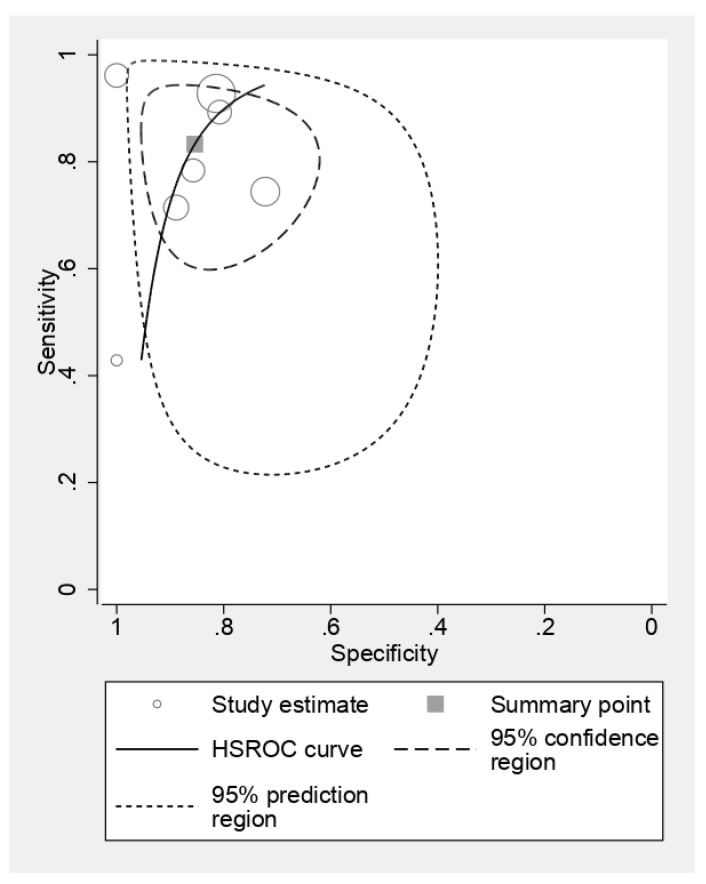
Standard summary receiver operating characteristic plot for MRI scan.

**Figure 7 cancers-16-01457-f007:**
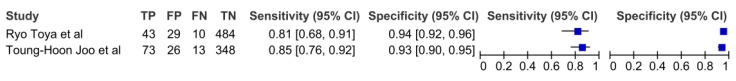
The forest plots of sensitivities and specificities of the PET–CT scan for diagnosis of ECS [38,43].

**Figure 8 cancers-16-01457-f008:**
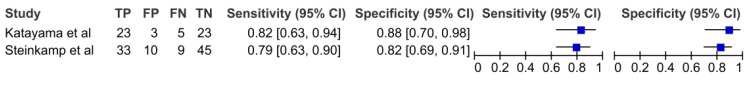
The forest plots of sensitivities and specificities of the USG scan for diagnosis of ECS [33,35].

**Table 1 cancers-16-01457-t001:** Demographic characteristics of the included studies.

Author	Year of Publication	Prospective/Retrospective	Number of Patients	Unit Considered	Mean Age	Male	Female	Site Included	HPV-Positive Only or Combined or Not Known	Imaging Modality Studied	Reference Standard	Threshold
Carvalho et al. [14]	1991	retrospective	28	Necks (31)	NK	NK	NK	NK	NK	CT	histology	capsule irregularity
Steinkamp et al. [25]	1999	retrospective	97	Patients	NK	NK	NK	NK	NK	CT, USG	histology	infiltration (CT) and capsule irregularity and necrosis (USG)
Steinkamp et al. [33]	2002	retrospective	79	Patients	NK	NK	NK	NK	NK	MRI	histology	necrosis
King et al. [20]	2004	retrospective	17	Nodes (51)	62.4	16	1	o, op, hp, l	NK	CT, MRI	histology	fat stranding and margin irregularity (CT) and necrosis (MRI)
Y. Kimura et al. [31]	2008	retrospective	109	Patients	66	89	20	o, op, np, hp, l	NK	MRI	histology	NK
Karaman et al. [27]	2009	retrospective	140	Patients		55	4	o, hp	NK	CT	histology	infiltration
Souter et al. [26]	2009	retrospective	127	Patients	NK	NK	NK	NK	NK	CT	histology	necrosis
Zoumalan et al. [34]	2010	retrospective	17	61 nodes	57	17	0	o, op, hp, l	NK	CT	histology	necrosis
Sumi et al. [30]	2011	retrospective	43	Nodes (54)	62	37	6	o, op, np, hp, l	NK	MRI	histology	capsule irregularity and infiltration
Katayama et al. [35]	2012	retrospective	50	Nodes (54)	NK	NK	NK	o, op, hp	NK	MRI, USG	histology	necrosis (MRI) and infiltration (USG)
C Url et al. [15]	2013	retrospective	49	Patients	60	44	5	o, op, hp, l	NK	CT	histology	infiltration
Lodder et al. [29]	2013	retrospective	39	Nodes (60)	63	24	15	o, op, hp, l	NK	MRI	histology	necrosis
Joo et al. [36]	2013	retrospective	78	106 level	NK	NK	NK	op	NK	PET	histology	SUV max 3.85
Joo et al. [37]	2013	retrospective	80	71 level	NK	NK	NK	o	NK	PET	histology	SUV max 2.25
R. Chai et al. [24]	2013	retrospective	100	Patients	62	79	21	o, op, l	NK	CT	histology	necrosis, fat stranding
young-Hoon Joo et al. [38]	2013	retrospective	57	Nodes (460)	61	55	2	hp	NK	PET	histology	SUV max 2.65
Prabhu et al. [23]	2014	retrospective	432	Patients	60	NK	NK	o, op, l	NK	CT	histology	infiltration
Aiken [11]	2015	prospective	111	Patients	NK	NK	NK	o	NK	CT	histology	necrosis (irregular borders, fat stranding, invasion)
Lee et al. [39]	2015	retrospective	263	Patients	NK	NK	NK	o, op, hp, l	NK	CT, PET	histology	necrosis (CT) SUV max 4.9 (PET)
Chun et al. [40]	2015	retrospective	89	Nodes (524)	NK	NK	NK	l	NK	PET	histology	SUV max 2.85
D Dequanter et al. [41]	2015	retrospective	54	Patients	NK	NK	NK	o, op, hp, l	NK	PET	histology	SUV max 4.15
L Maxwell et al. [21]	2015	retrospective	65	Patients	55.9	60	5		Combined	CT	histology	margin irregularity
Randall et al. [42]	2015	retrospective	40	77 nodes	NK	29	11	o	NK	CT	histology	necrosis
J Liu et al. [19]	2016	retrospective	96	Patients	58	116	24	o, op, np, hp, l	NK	CT	histology	(thick wall, enhancing margin, loss of nodal margin, infiltration)
A Sharma et al. [13]	2017	prospective	30	Patients	52.9	24	6	o	NK	CT	histology	central necrosis
Geltzeiler [17]	2017	prospective	100	Patients	NK	NK	NK	op	HPV +ve	CT	histology	infiltration, matte nodes
J Carlton et al. [18]	2017	prospective	93	Patients	61	58	25	o, op, np, l	NK	CT	histology	central necrosis
K Moreno et al. [28]	2017	prospective	20	Neck (34) and nodes (12)	58	12	8	o	NK	MRI	histology	NK
M Patel et al. [22]	2018	prospective	27	Patients	57	27	0	op, np	HPV +ve	CT	histology	necrosis, lobular pattern (perinodal stranding, matted appearance, invasion)
A Noor et al. [12]	2019	prospective	80	Nodes (91)	58	68	12	op	NK	CT	histology	perinodal fat stranding (capsule contour, invasion)
Ryo Toya et al. [43]	2020	retrospective	94	Nodes (566)	NK	NK	NK	o, op, hp, l	NK	PET	histology	SUV max 2.3
Sheppard et al. [44]	2020	retrospective	176	Patients	NK	NK	NK	o, op, hp, l	NK	PET	histology	SUV max 10
Sheppard et al. [44]	2020	retrospective	166	Patients	NK	NK	NK	o, op, hp, l	NK	MRI	histology	margin irregularity

NK: not known.

**Table 2 cancers-16-01457-t002:** Pooled diagnostic accuracy of various imaging modalities in detecting extracapsular spread.

Imaging	Pooled Sensitivity (95% Confidence Interval)	Pooled Specificity (95% Confidence Interval)	Likelihood Ratios +ve (95% Confidence Interval)	Likelihood Ratios −ve (95% Confidence Interval)	Diagnostic Odds Ratio (95% Confidence Interval)
CT	0.63 [95% CI = 0.53–0.73]	0.85 [95% CI = 0.74–0.91]	4.33 [95% CI = 2.63–7.05]	0.425 [95% CI = 0.33–0.53].	10.1 [95% CI = 5.89–17.42]
MRI	0.83 [95% CI = 0.71–0.90]	0.85 [95% CI = 0.73–0.92]	5.7 [95% CI = 2.96–10.99]	0.19 [95% CI = 0.10–0.35]	29.18 [95% CI = 9.79–86.94]
PET	0.80 [95% CI = 0.74–0.85]	0.93 [95% CI = 0.92–0.94]	12.3 [95% CI = 9.9–15.26]	0.21 [95% CI = 0.15–0.29].	57.75 [95% CI = 37.37–89.25]
USG	0.80 [95% CI = 0.68–0.88]	0.84 [95% CI = 0.74–0.91]	4.83 [95% CI = 2.89–8.07]	0.23 [95% CI = 0.14–0.38]	20.69 [95% CI = 8.9–48.08]

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
