# Peer review of "A Systematic Review and Meta-Analysis of 29 Studies Predicting Diagnostic Accuracy of CT, MRI, PET, and USG in Detecting Extracapsular Spread in Head and Neck Cancers"

_cancers, 2024, doi:10.3390/cancers16081457_

Round 1

Reviewer 1 Report

Comments and Suggestions for Authors

The authors presented a paper entitled "A Systematic Review and Meta-Analysis of 25 Studies Predicting Diagnostic Accuracy of CT, MRI, PET and USG in Detecting Extracapsular Spread in Head and Neck Cancers".

The topic is interesting and the authors have certainly made a huge effort in collecting data however I would suggest in order to improve the quality of the manuscript to perform a few additions as follows:

1) Please modify figure 1 removing the "additional records" box which provided 0 results and which is not even mentioned in the search strategy

2) In figure 1 please list the number of papers excluded after eligibility assessment according to each specific reason (not merely recalling the use of inclusion and exclusion criteria)

3) Why did not the authors include a relevant databse such as Scopus to perfomr their search? this could lead to some relvant papers being missed; please consider to extend the search also to this database

4) Why did the authors choose to limit the search strategy to June 2021? From June 2021 to January 2024 there are 30 months of potential useful papers which could be missed; please extend the seaxrh at least to December 2023

Author Response

Reviewer 1 comments and response:

  • Please modify figure 1 removing the "additional records" box which provided 0 results and which is not even mentioned in the search strategy

1R: Thank you for the suggestion. Mmodification has been done accordingly in figure 1.

  • In figure 1 please list the number of papers excluded after eligibility assessment according to each specific reason (not merely recalling the use of inclusion and exclusion criteria)

2R:144 articles has been excluded according to exclusion criteria including 1. Missing data on imaging characteristics of ECS. 2.Inaccurate definition of extra capsular extension.3. Missing data on lymph node status.

  • Why did not the authors include a relevant databasesuch as Scopus to perform their search? this could lead to some relevant papers being missed; please consider to extend the search also to this database

3R:  Scopus indexed articles are limited availability, poor coverage of books and conference proceedings, and inadequate mechanisms for distinguishing among authors. Moreover, as per the AMSTER guidelines, atleast two database are needed for SR/MA. We have included three databases which include PUBMED that provides Easy keyword searching and automatic mapping to MeSH terms.

  • Why did the authors choose to limit the search strategy to June 2021? From June 2021 to January 2024 there are 30 months of potential useful papers which could be missed; please extend the searchat least to December 2023

4R: We have extended our search till December 2023 and found one article relevant (review article) which is excluded because it doesn’t fit with the inclusion criteria. We did search till December and no additional articles were found based on our inclusion criteria

Reviewer 2 Report

Comments and Suggestions for Authors

This manuscript presents a systematic review and meta-analysis of 25 studies predicting diagnostic accuracy of various image modalities in detecting extracapsular spread in head and neck cancers. 

Major comments

 Abstract

--

 Introduction

--

 Material and Methods

 Including studies from 1990 on may provide substantial bias, because of inferior technology of old generation scanners. Please discuss. 

The title indicated 25 studies, however, the PRISMA fow diagram in Figure 1 indicates n=29 studies. Please correct or explain. 

Why did you exclude the study of Url C, Schartinger VH, Riechelmann H, Glückert R, Maier H, Trumpp M, Widmann G. Radiological detection of extracapsular spread in head and neck squamous cell carcinoma (HNSCC) cervical metastases. Eur J Radiol. 2013 Oct;82(10):1783-7. doi: 10.1016/j.ejrad.2013.04.024. Epub 2013 Jun 7. PMID: 23751931.

 Discussion:

--

Besides image modality, use of the correct criteria for extracapsular spread seems more important (e.g. apparent fat and soft tissue infiltration or infiltration of sternocleidomastoid muscle, internal jugular vein or carotid artery). Please discuss and review the image criteria for ECS in the selected 29 studies in a more comprehensive way.  

Please provide a list of image criteria and their diagnostic accuracy in all the different image modalities.

Comments on the Quality of English Language

-

Author Response

Reviewers 2 comments and response:

 Material and Methods

  1. Including studies from 1990 on may provide substantial bias, because of inferior technology of old generation scanners. Please discuss.

1R: As per the suggestion of reviewer we have added the above suggestion as a limitation in our manuscript.

  1. The title indicated 25 studies, however, the PRISMA flow diagram in Figure 1 indicates n=29 studies. Please correct or explain.

2R: Thank you for the vital information. Modification has been accordingly.

  1. Why did you exclude the study of Url C, Schartinger VH, Riechelmann H, Glückert R, Maier H, Trumpp M, Widmann G. Radiological detection of extracapsular spread in head and neck squamous cell carcinoma (HNSCC) cervical metastases. Eur J Radiol. 2013 Oct;82(10):1783-7. doi: 10.1016/j.ejrad.2013.04.024. Epub 2013 Jun 7. PMID: 23751931

3r: Its included as our 12th reference and also its part of our systematic review and analysis (Table 1)

4. Besides image modality, use of the correct criteria for extracapsular spread seems more important (e.g. apparent fat and soft tissue infiltration or infiltration of sternocleidomastoid muscle, internal jugular vein or carotid artery). Please discuss and review the image criteria for ECS in the selected 29 studies in a more comprehensive way. 

4R: Thank you for the inputs. Criteria for ECS (for the included studies) has been added in table 1.

5. Please provide a list of image criteria and their diagnostic accuracy in all the different image modalities.

5R: Thank you for the inputs. Criteria for ECS (for the included studies) has been added in table 1. The diagnostic accuracy in each study can be identified in the forest plot and image criteria has been added in the demographic table

These two are for the review 2.

Reviewer 3 Report

Comments and Suggestions for Authors

The authors assessed the diagnostic accuracy of currently available imaging modalities to detect the presence of extracapsular extension (ECS) in head and neck cancer (HNC), specifically CT, MRI, PET-CT, and ultrasound. It has been shown that although MRI and PET-CT are more sensitive than CT for assessing the presence of pacemaker, the increase in overall diagnostic accuracy does not appear to be statistically significant. All imaging modalities had similar specificity in detecting ECS.

1. In Table 1, I would recommend that authors group studies by imaging method.

2. The abstract indicates that the study included 34 papers, while Figure 1 shows 29. Which is correct?

3. Figure 5 - there is a typo in the caption: is CT or MRI still correct?

4. Is there a difference between using contrast and without it for CT?

5. Will the diagnostic accuracy differ for different lymph nodes: level I and IIA lymph nodes; level IIB-VI; retropharyngeal?

Author Response

Reviewers 3 comments and response:

The authors assessed the diagnostic accuracy of currently available imaging modalities to detect the presence of extra capsular extension (ECS) in head and neck cancer (HNC), specifically CT, MRI, PET-CT, and ultrasound. It has been shown that although MRI and PET-CT are more sensitive than CT for assessing the presence of pacemaker, the increase in overall diagnostic accuracy does not appear to be statistically significant. All imaging modalities had similar specificity in detecting ECS.

  1. In Table 1, I would recommend that authors group studies by imaging method.

1R: Thank you for the suggestion. We have mmodified in table 1.

  1. The abstract indicates that the study included 34 papers, while Figure 1 shows 29. Which is correct?

2r:  Thank you for the suggestion. Manuscript has been modified.

  1. Figure 5 - there is a typo in the caption: is CT or MRI still correct?

3R: Thank you for pointing out. Correction has been done.

  1. Is there a difference between using contrast and without it for CT?

4R: Thank you. This information has been added to the table 1.

  1. Will the diagnostic accuracy differ for different lymph nodes: level I and IIA lymph nodes; level IIB-VI; retropharyngeal?

5R: This is an interesting topic but to answer this question, we need a separate metanalyses and this would be beyond the scope of this study

Reviewer 4 Report

Comments and Suggestions for Authors

This study investigates the effectiveness of different imaging techniques, including CT, magnetic resonance imaging, positive emission tomography and ultrasonography, in detecting extracapsular spread in advanced head and neck cancer. The obtained results suggested the need to refine the criteria used in imaging to better diagnose extracapsular spread in head and neck cancer, which could significantly impact how this cancer is treated in the future. The manuscript is well prepared. I may suggest publication of this work in Cancers. Although the manuscript is scientifically Ok, it is not prepared in well understandable ways for non-specialist readers. Therefore, I suggest several revisions on general points.

1) This manuscript is visually prepared with flow chart, tables, and conceptual graphs. Addition of some explanation figure for cancers and analyses would attract non-specialist readers.

2) Similarly, one figure to represent conclusion would be helpful.

3) At the last part in conclusion, future impact, perspective, and problems can be more concretely described in more details.

Author Response

Reviewers 4 comments and response:

This study investigates the effectiveness of different imaging techniques, including CT, magnetic resonance imaging, positive emission tomography and ultrasonography, in detecting extracapsular spread in advanced head and neck cancer. The obtained results suggested the need to refine the criteria used in imaging to better diagnose extracapsular spread in head and neck cancer, which could significantly impact how this cancer is treated in the future. The manuscript is well prepared. I may suggest publication of this work in Cancers. Although the manuscript is scientifically Ok, it is not prepared in well understandable ways for non-specialist readers. Therefore, I suggest several revisions on general points.

  • This manuscript is visually prepared with flow chart, tables, and conceptual graphs. Addition of some explanation figure for cancers and analyses would attract non-specialist readers.

1R: Thank you for suggestion. There are 7 figures and 2 tables with detailed legend submitted.

2)Similarly, one figure to represent conclusion would be helpful.

2R: Thank you for the suggestion. Table 2 concludes the comparative impact of various imaging techniques on detecting ECS.

3)At the last part in conclusion, future impact, perspective, and problems can be more concretely described in more details.

3R: Thank you the comments. It has been imbibed in the manuscript.

Round 2

Reviewer 1 Report

Comments and Suggestions for Authors

The authors did not adequately consider my previous comments: please provide the reason in the figure and in the text for excluding some papers (not only in the reply to the reviewer) and expand your search incliding Scopus which can not be be simply excluded for the reasons provided by the authors.

Author Response

Scopus search was extended and we didnt find any article fitting into our inclusion criteria apart from already included in the study.

We have provided reason for exclusion in the PRISMA diagram.

Reviewer 2 Report

Comments and Suggestions for Authors

—-

Author Response

Thank you for your comments and suggestions

Reviewer 3 Report

Comments and Suggestions for Authors

I have no further comments on the article.

Author Response

Thank you for your suggestions and comments

Round 3

Reviewer 1 Report

Comments and Suggestions for Authors

No further comments